# Diffusion Priors In Variational Autoencoders

**Antoine Wehenkel** [1]   **Gilles Louppe** [1]

## Abstract

Among likelihood-based approaches for deep generative modelling, variational autoencoders (VAEs) offer scalable amortized posterior inference and fast sampling. However, VAEs are also more and more outperformed by competing models such as normalizing flows (NFs), deep-energy models, or the new denoising diffusion probabilistic models (DDPMs). In this preliminary work, we improve VAEs by demonstrating how DDPMs can be used for modelling the prior distribution of the latent variables. The diffusion prior model improves upon Gaussian priors of classical VAEs and is competitive with NF-based priors. Finally, we hypothesize that hierarchical VAEs could similarly benefit from the enhanced capacity of diffusion priors.

## 1. Introduction

Over the last few years, the interest of the deep learning community for generative modelling has increased steadily. Among the likelihood-based approaches for deep generative modelling, variational autoencoders (Kingma & Welling, 2013, VAEs) stand as one of the most popular, although competing approaches now demonstrate better performance. In particular, Ho et al. (2020); Nichol & Dhariwal (2021); Dhariwal & Nichol (2021) recently showed that denoising diffusion probabilistic models (DDPMs) are competitive deep generative models, obtaining samples quality similar to those of the best implicit deep generative models such as ProgressiveGAN (Karras et al., 2017) and Style-GAN (Karras et al., 2019). Similarly to VAEs, DDPMs train on a variational bound and may be interpreted under the encoding-decoding framework.

In the original formulation of VAEs, the prior and the posterior distributions over the latent variables are assumed to

---

*Equal contribution   [1]University of Liège, Liège, Belgium. Correspondence to: Antoine Wehenkel .

Third workshop on *Invertible Neural Networks, Normalizing Flows, and Explicit Likelihood Models* (ICML 2021).

be both Gaussian. However, these assumptions are often incompatible and are thus limiting the performance of VAEs for complex modelling tasks (Tomczak & Welling, 2018; Chen et al., 2018). A natural solution to this problem is to parameterize the prior, sometimes also the posterior, with more expressive distributions. In this preliminary work, we improve VAEs by demonstrating how DDPMs can be used for modelling the prior distribution of the latent variables. In addition to boosting DDPM with the compression properties of VAEs, combining the two models should eventually lead to greater generative performance by enabling complex generative modelling even with simple decoder architecture. Finally, working in the latent space should eventually reduce the computational burden associated with diffusion generative models.

## 2. Latent generative models

### 2.1. Variational autoencoder

We want to learn a generative model of an unknown distribution $p(\mathbf{x})$ given a dataset $X \in \mathbb{R}^{N \times d}$ of $N$ i.i.d observations $\mathbf{x}$ sampled from this unknown distribution. The original VAE postulates a two-step generative process in which some unobserved variables $\mathbf{z} \in \mathbb{R}^h$ are first sampled from a prior distribution $p(\mathbf{z})$ and then observations $\mathbf{x}$ are generated from a conditional distribution $p_\theta(\mathbf{x}|\mathbf{z})$. The generative process can be expressed mathematically as

$$\mathbf{z} \sim p(\mathbf{z}) \quad \text{and} \quad \mathbf{x} \sim p_\theta(\mathbf{x}|\mathbf{z}). \tag{1}$$

The prior $p(\mathbf{z})$ is chosen Gaussian while the likelihood $p_\theta(\mathbf{x}|\mathbf{z})$ is modeled with a neural network. The likelihood model decodes latent variables into observations and is thus usually refereed as the decoder in the literature. In its original formulation, the likelihood is parameterized with a multivariate Gaussian $\mathcal{N}(\mu_\theta(\mathbf{z}), \text{diag}(\sigma_\theta^2(\mathbf{z})))$ when the observations are continuous, and a categorical distribution when they are discrete.

Training the generative model is achieved by finding the parameters $\theta$ of the decoder that maximize the sum of the marginal likelihoods of individual points

$$p_\theta(X) = \sum_{\mathbf{x} \in X} \log \int p_\theta(\mathbf{x}|\mathbf{z})p(\mathbf{z})\mathrm{d}\mathbf{z}.$$

These integrals are intractable but the introduction of an

encoder network that approximates the posterior distribution $q_\phi(\mathbf{z}|\mathbf{x})$ allows maximizing the associated evidence lower bound

$$\text{ELBO} := \mathbb{E}_q \left[ \log \frac{p_\theta(\mathbf{x}|\mathbf{z})p(\mathbf{z})}{q_\psi(\mathbf{z}|\mathbf{x})} \right] \qquad (2)$$

$$= \log p_\theta(\mathbf{x}) - \mathbb{KL}\left[ q_\psi(\mathbf{z}|\mathbf{x}) || p_\theta(\mathbf{z}|\mathbf{x}) \right] \qquad (3)$$

$$\leq \log p_\theta(\mathbf{x}). \qquad (4)$$

The ELBO becomes tighter as the approximate posterior $q_\psi(\mathbf{z}|\mathbf{x})$ gets closer to the true posterior. Learning the generative model is finally performed by jointly optimizing the parameters $\theta$ of the decoder and $\phi$ of the approximate posterior via stochastic gradient ascent. In the original VAE, the encoder models the approximate posterior as a conditional multivariate Gaussian distribution $\mathcal{N}(\mu_\phi(\mathbf{x}), \text{diag}(\sigma_\phi^2(\mathbf{x})))$.

The ELBO loss presents two antagonistic goals to the encoder. It should be able to both encodes the data accurately while being as close as possible to the prior distribution. Consequently, the Gaussian assumptions made on both the prior and the posterior distributions are often incompatible and limit the generative performance. A possible solution consists in learning a prior distribution that is compatible with the learned posteriors. For example, Habibian et al. (2019) and Chen et al. (2017) respectively showed that autoregressive models and normalizing flows (Rezende & Mohamed, 2015, NFs) greatly improve the performance of VAEs when used as prior distributions. In the following we present how denoising diffusion probabilistic models can be used to improve the performance of classical VAEs.

## 2.2. Denoising diffusion probabilistic models

Inspired by non-equilibrium statistical physics, Sohl-Dickstein et al. (2015) originally introduced DDPMs while Ho et al. (2020) demonstrated only more recently how to train these models for image synthesis, achieving results close to the state-of-the-art on this task. DDPMs formulate generative modelling as the reverse operation of diffusion, a physical process which progressively destroys information. Formally, the reverse process is a latent variable model of the form

$$p_\phi(\mathbf{x}_0) := \int p_\phi(\mathbf{x}_{0:T}) d\mathbf{x}_{1:T},$$

where $\mathbf{x}_0 := \mathbf{x}$ denotes the observations and $\mathbf{x}_1, \ldots, \mathbf{x}_T$ denote latent variables of the same dimensionality as $\mathbf{x}_0$. The joint distribution $p_\phi(\mathbf{x}_{0:T})$ is modelled as a first order Markov chain with Gaussian transitions, that is

$$p_\phi(\mathbf{x}_{0:T}) := p_\phi(\mathbf{x}_T) \prod_{t=1}^{T} p_\phi(\mathbf{x}_{t-1}|\mathbf{x}_t), \qquad (5)$$

$$p_\phi(\mathbf{x}_T) := \mathcal{N}(\mathbf{0}, \mathbf{I}), \qquad (6)$$

$$p_\phi(\mathbf{x}_{t-1}|\mathbf{x}_t) := \mathcal{N}(\mu_\phi(\mathbf{x}_t, t), \sigma_t^2 \mathbf{I}). \qquad (7)$$

Similar to VAEs, the reverse Markov chain is trained on an ELBO. However, the approximate posterior $q(\mathbf{x}_{1:T}|\mathbf{x}_0)$ is fixed to a diffusion process that is also a first order Markov chain with Gaussian transitions,

$$q(\mathbf{x}_{1:T}|\mathbf{x}_0) := \prod_{t=1}^{T} q(\mathbf{x}_t|\mathbf{x}_{t-1}), \qquad (8)$$

$$q(\mathbf{x}_t|\mathbf{x}_{t-1}) := \mathcal{N}(\sqrt{1-\beta_t}\mathbf{x}_{t-1}, \beta_t \mathbf{I}), \qquad (9)$$

where $\beta_1, \ldots, \beta_T$ are the variance schedule that is either fixed as training hyper-parameters or learned. The ELBO is then given by

$$\text{ELBO} := \mathbb{E}_q \left[ \log \frac{p_\phi(\mathbf{x}_{0:T})}{q(\mathbf{x}_{1:T}|\mathbf{x}_0)} \right] \leq \log p_\phi(\mathbf{x}_0). \qquad (10)$$

Provided that the variance schedule $\beta_t$ is small and that the number of timesteps $T$ is large enough, the Gaussian assumptions on the generative process $p_\phi$ are reasonable. Ho et al. (2020) take advantage of the Gaussian transitions to express the ELBO as

$$\mathbb{E}_q \Bigg[ \mathbb{KL}\left[ q(\mathbf{x}_T|\mathbf{x}_0) || p(\mathbf{x}_T) \right] - \log p_\phi(\mathbf{x}_0|\mathbf{x}_1)$$
$$+ \sum_{t=2}^{T} \mathbb{KL}\left[ q(\mathbf{x}_{t-1}|\mathbf{x}_t, \mathbf{x}_0) || p_\phi(\mathbf{x}_{t-1}|\mathbf{x}_t) \right] \Bigg]. \qquad (11)$$

The inner sum in Equation (11) is made of comparisons between the Gaussian generative transitions $p_\phi(\mathbf{x}_{t-1}|\mathbf{x}_t)$ and the conditional forward posterior $q(\mathbf{x}_{t-1}|\mathbf{x}_t, \mathbf{x}_0)$ which can also be expressed in closed form as Gaussians $\mathcal{N}(\tilde{\mu}_t(\mathbf{x}_0, \mathbf{x}_t), \tilde{\beta}_t \mathbf{I})$, where $\tilde{\beta}_t$ are functions of the variance schedule. The KL can thus be calculated with closed form expressions which reduces the variance of the final expression. In addition, Ho et al. (2020) empirically demonstrate that it is sufficient to take optimization steps on uniformly sampled terms of the sum instead of computing it completely. The final objective closely resembles denoising score matching over multiple noise levels (Song & Ermon, 2019). These observations combined with additional simplifications leads to a simplified loss

$$L_{\text{DDPM}}(\mathbf{x}_0; \phi) := \mathbb{E}_{t, \mathbf{x}_0, \mathbf{x}_t} \left[ \frac{1}{2\sigma_t^2} ||\mu_\phi(\mathbf{x}_t, t) - \tilde{\mu}_t(\mathbf{x}_0, \mathbf{x}_t)||^2 \right],$$
$$(12)$$

where $\tilde{\mu}_t(\mathbf{x}_0, \mathbf{x}_t)$ is the mean of $q(\mathbf{x}_{t-1}|\mathbf{x}_0, \mathbf{x}_t)$, the forward diffusion posterior conditioned on the observation $\mathbf{x}_0$.

## 3. Prior modelling with denoising diffusion

We now introduce our contribution which consists in using a DDPM for modelling the prior distribution in VAEs. We

formulate the generative model as

$$\mathbf{z}_T \sim \mathcal{N}(\mathbf{0}, \mathrm{I}) \tag{13}$$

$$\mathbf{z}_{t-1|t} \sim p_\phi(\mathbf{z}_{t-1}|\mathbf{z}_t) \quad \forall t \in [T, \dots, 1] \tag{14}$$

$$\mathbf{x} \sim p_\theta(\mathbf{x}|\mathbf{z}_0), \tag{15}$$

where $\phi$ denotes the parameters of the reverse diffusion model encoding the prior distribution. Equations (13) and (14) implicitly define a prior distribution over the usual latent variables $\mathbf{z}_0$ which is modelled with a reverse diffusion process.

Unfortunately, we cannot train a VAE with a diffusion prior directly on the ELBO as expressed in Equation (2) as $p_\phi(\mathbf{z}_0)$ cannot be evaluated. However, Equation (2) can be further developed as

$$\mathbb{E}_{q_\psi}\left[\log p_\theta(\mathbf{x}|\mathbf{z}_0)\right] - \mathbb{E}_{q_\psi}\left[\log q(\mathbf{z}_0|\mathbf{x})\right] + \mathbb{E}_{q_\psi}\left[\log p_\phi(\mathbf{z}_0)\right] \tag{16}$$

in which a lower bound on the last term can be expressed by Equation (10). This finally leads to the following expression

$$\mathbb{E}_{q_\psi}\left[\log p_\theta(\mathbf{x}|\mathbf{z}_0) - \log q(\mathbf{z}_0|\mathbf{x}) + \mathbb{E}_q\left[\log \frac{p_\phi(\mathbf{z}_{0:T})}{q(\mathbf{z}_{1:T}|\mathbf{z}_0)}\right]\right] \tag{17}$$

$$\leq \mathbb{E}_{q_\psi}\left[\log p_\theta(\mathbf{x}|\mathbf{z}_0) - \log q(\mathbf{z}_0|\mathbf{x}) + \log p_\phi(\mathbf{z}_0)\right] \tag{18}$$

$$\leq \log p_\theta(\mathbf{x}), \tag{19}$$

which is a valid ELBO. Finally, the diffusion prior $p_\phi$ is trained jointly with the approximate posterior $q_\psi$ and the likelihood models $p_\theta$ which are optimized as in a classical VAE. This leads to the following loss function:

$$\mathcal{L}(\mathbf{x}; \phi, \theta, \psi) := \mathbb{E}_{q_\psi}\left[\log \frac{p_\theta(\mathbf{x}|\mathbf{z})}{q_\psi(\mathbf{z}|\mathbf{x})}\right] + \mathbb{E}_{q_\psi}\left[L_{\text{DDPM}}(\mathbf{z}_0; \phi)\right]. \tag{20}$$

## 4. Related work

Various approaches have been proposed to improve the modelling capacity and the training of VAEs. As a first example, some state-of-the-art deep generative models based on VAEs model the posterior with normalizing flows or autoregressive models (Kingma et al., 2016; Vahdat & Kautz, 2020). Autoregressive models are also often used as a replacement of the original likelihood parameterization, which assumes conditional independencies that are often unrealistic (Oord et al., 2016). Another popular improvement made to the original VAE is the embedding of structure in the latent variables. In particular, hierarchical VAEs (Sønderby et al., 2016; Kingma et al., 2016) combined with careful training demonstrate impressive results on generative modelling for images (Vahdat & Kautz, 2020).

Vahdat et al. (2021) concurrently proposed to use diffusion for modelling the prior distributions of VAEs. They obtain

state-of-the-art results on image synthesis by combining continuous diffusion models and VAEs. Not as close to our work but related, Chen et al. (2017) proposed to learn the prior as a solution to the mismatch between the approximate and the true posteriors. They model the prior with an autoregressive flow, which also closely relates to modelling the posterior distribution with an inverse autoregressive flow (Kingma et al., 2016). Tomczak & Welling (2018) takes inspiration from the aggregated posterior $\frac{1}{N}\sum_{i=1}^{N}q_\psi(z|x)$ (Hoffman & Johnson, 2016; Makhzani et al., 2015) to introduce the VampPrior defined as a mixture of learned pseudo-inputs. An orthogonal line of work suggests that the mismatch between the approximate posterior and the exact posterior can be reduced by over-weighting the terms related to the prior and to the approximate posterior in the ELBO (Higgins et al., 2016; Chen et al., 2018).

## 5. Experiments

We now compare VAEs for different choices of priors, including the original Gaussian prior, an NF prior, and the proposed diffusion prior. All models share a same backbone encoder-decoder architecture inspired from DCGAN (Radford et al., 2015). Optimization is performed with Adam for 250 epochs with a learning rate set to 0.0005. After each epoch, the models are evaluated on a validation set used to select the best one for each training setting. We compare the models on the CIFAR10 and CelebA datasets for 3 different latent variables dimensionality (40, 100, 200). The NF used in our experiments is a 3-step autoregressive affine flow with simple MLP backbones similar to the one used to model the transition function of DDPM.

Table 1 presents the FID scores for the different models. We first notice the large scores reached by all models on the CIFAR10 dataset. This can be explained by the simplicity of the models trained in our experiments. We believe these scores could be greatly improved by using a more sophisticated likelihood model such as a PixelCNN (Oord et al., 2016). Although FID scores suggest that the Gaussian prior outperforms the diffusion prior in terms of generative performance, the visual inspection of Figure 1 shows that the diffusion prior results in samples slightly more realistic than those of the classical VAE. The best FID score is achieved by the NF prior, although its samples do not seem to reflect this superiority. In this case, we believe the FID scores are not entirely informative about the quality of the images synthesized by the models and should be interpreted with a grain of salt. Although learned priors seem to improve generative performance on CIFAR10, additional work is needed to reach results that would justify using a diffusion prior for this dataset.

On CelebA however, we observe in Table 1 that diffusion priors outperform the Gaussian prior. This is in line with

*Table 1.* FID scores for different models for prior modelling in VAEs and for different latent size. *Diffusion priors outperform classical VAE on CelebA but are slightly worse than NFs. FID scores do not reveal the superiority of any method on CIFAR10.*

| Dataset | CelebA | | | CIFAR10 | | |
|---|---|---|---|---|---|---|
| Latent Size | 40 | 100 | 200 | 40 | 100 | 200 |
| Gaussian | 154.3 | 149.4 | 139.1 | 176.0 | 126.2 | 123.9 |
| NF | 72.9 | 59.49 | 54.7 | 167.6 | 129.1 | 129.6 |
| Diffusion | 114.8 | 67.95 | 88.3 | 177.9 | 160.5 | 153.1 |

the visual inspection of Figure 2a and Figure 2c. As for CIFAR10, the NF prior outperforms the Gaussian and diffusion priors in terms of FID scores, although the visual inspection of the corresponding samples in Figure 2b does not reveal a much better quality of images when compared to those resulting from the diffusion prior. We conclude from these observations that diffusion priors offer an interesting alternative to NFs for modelling the prior in a VAE.

## 6. Conclusion and future work

This preliminary work presents how denoising diffusion probabilistic models can be used as a new class of learnable priors for VAEs. As a notable contribution, we empirically demonstrate that optimizing implicitly a prior on an ELBO can be performed jointly to training the encoder and the decoder of the VAE. In addition, our results suggest DDPM performs on par with NFs for modelling prior distribution.

A large spectrum of future research directions could benefit from the basic idea expressed in this preliminary work. As an example, recent advances in diffusion models such as the continuous formulation (Song et al., 2020) or improvement to the training procedure of DDPM (Nichol & Dhariwal, 2021) could be implemented in the prior model. Similarly, many improvements could be made to the architectures used for the VAE and to the training procedure. In particular, image synthesis with hierarchical VAEs which organizes the latent variables into multiple scales images could reveal the full potential of diffusion priors. This would indeed combine the structural knowledge embed by such type of VAEs with the impressive performance of DDPM for modelling distributions over images. Finally, diffusion does not constrain the neural networks architectures and so enables the embedding of a larger choice of inductive biases in the prior distribution compared to autoregressive models and NFs.

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

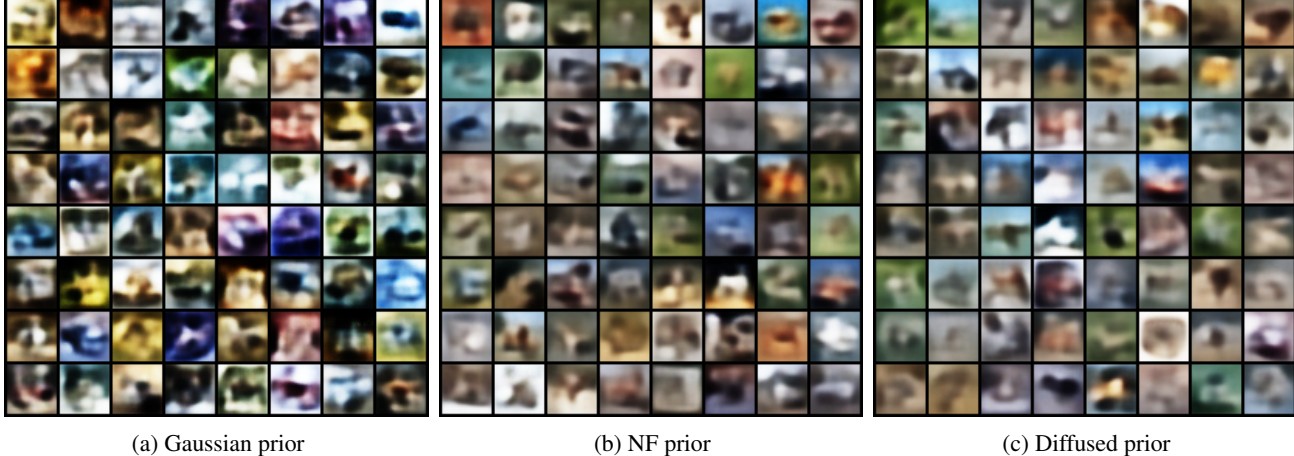

(a) Gaussian prior      (b) NF prior      (c) Diffused prior

*Figure 1.* Samples generated by a VAE trained on CIFAR10 for three different prior models. *The diffusion prior leads to slightly better sampling quality than the Gaussian distribution and similar to the NF prior.*

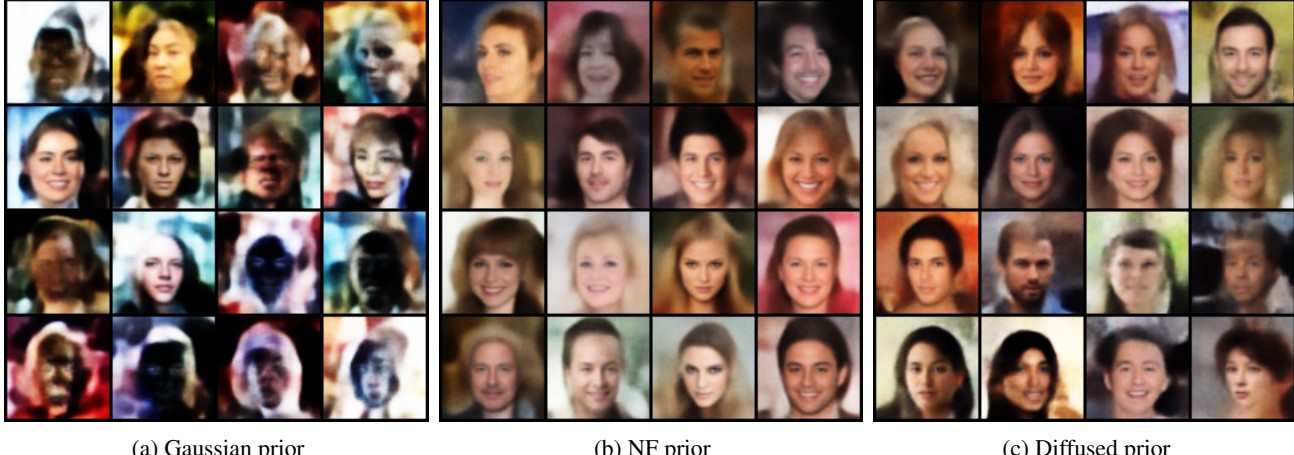

(a) Gaussian prior      (b) NF prior      (c) Diffused prior

*Figure 2.* Samples generated by a VAE trained on CelebA for three different prior models. *The diffusion prior leads to better sampling quality than the Gaussian distribution and similar to the NF prior.*

Rezende, D. and Mohamed, S. Variational inference with normalizing flows. In *International Conference on Machine Learning*, pp. 1530–1538. PMLR, 2015.

Sohl-Dickstein, J., Weiss, E., Maheswaranathan, N., and Ganguli, S. Deep unsupervised learning using nonequilibrium thermodynamics. In *International Conference on Machine Learning*, pp. 2256–2265. PMLR, 2015.

Sønderby, C. K., Raiko, T., Maaløe, L., Sønderby, S. K., and Winther, O. Ladder variational autoencoders. *arXiv preprint arXiv:1602.02282*, 2016.

Song, Y. and Ermon, S. Generative modeling by estimating gradients of the data distribution. In *Proceedings of the 33rd Annual Conference on Neural Information Processing Systems*, 2019.

Song, Y., Sohl-Dickstein, J., Kingma, D. P., Kumar, A., Ermon, S., and Poole, B. Score-based generative modeling through stochastic differential equations. *arXiv preprint arXiv:2011.13456*, 2020.

Tomczak, J. and Welling, M. Vae with a vampprior. In *International Conference on Artificial Intelligence and Statistics*, pp. 1214–1223. PMLR, 2018.

Vahdat, A. and Kautz, J. Nvae: A deep hierarchical variational autoencoder. *arXiv preprint arXiv:2007.03898*, 2020.

Vahdat, A., Kreis, K., and Kautz, J. Score-based generative modeling in latent space. *arXiv preprint arXiv:2106.05931*, 2021.