# OpenReview forum: "Diffusion Priors In Variational Autoencoders"
_ICML.cc/2021/Workshop/INNF — INNF+ 2021 poster_

### Official Review · Reviewer_UxDg · 2021-06-07

**Rating:** Borderline Accept
**Confidence:** 4

**Summary:**

Summary: This paper introduces diffusion models as a prior in VAEs. The paper shows qualitative results by showing samples, and quantitative results via FID scores, which are not entirely conclusive on which model is better.

As the authors themselves describe, the proposed model is a logical step coming from VAE + NF prior literature, and instead using diffusion. The paper is well-written and the exposition of the background in section 2 is thorough and I think very helpful as an introduction to the subjects. The main weakness of the paper is that the idea might be a little straightforward, and then empirically is missing some important details: When a diffusion model is trained using Eq.  (10), there is 1 term p(x0 | x1) that can become sharp/peaked when it is a continuous distribution. This problem does not occur in DDPMs as this final p(x0|x1) term is discrete. This problem is alleviated by training with Eq. (12) but the log-likelihoods may be very poor as this last equation is not properly calibrated. It would be interesting to investigate these issues of diffusion models for continuous distributions further. This is then also related to the main weakness of the experimental section: there are in the end no reported ELBO values. Even if these ELBO values show much worse performance for VAEs + DDPMs compared to VAE or VAE + NF, I would still like to ask the authors to put them in. Perhaps the above discussion on continuous diffusion might help explain the mismatch in performance.

Pros:
- Straightforward logical step, might be very interesting for hybrids between VAEs and Diffusion
- Very well-written exposition of background

Cons:
- Missing discussion of issues with a continuous p(x0 | x1)
- Missing empirical ELBO evaluation

Some suggestions/minor comments:
- L72 right column. Is this formulation not only from Ho et al, but also from Sohl-Dickstein et al?
- Section 2.3. I would change this to a main section, so "Section 3", since this is your main method
- Section 2 in general: I would perhaps rephrase the background in terms of variables z_0:T for diffusion, as this connects better to Eqs. (13-20) in your method. Otherwise the shift in meaning between x and z might be confusing to readers.


**Justification For Rating:**

The main reasons for a borderline accept is that ELBO evaluation is missing and I think that diffusion for continuous distributions could have been explored further. See above for details.

---

### Official Review · Reviewer_Vj7M · 2021-06-08

**Rating:** Accept
**Confidence:** 4

**Summary:**

The paper considers the problem inconsistency between the learned posterior and a prior distribution. The authors address the problem by learning a prior distribution based on denoising diffusion probabilistic models along with learning the likelihood and the approximate posterior.

**Justification For Rating:**

The motivation is clear, and the proposed framework is well-justified and described in a concise way. Overall, the paper is well-written and easy to follow. As a comment, it would be helpful to see some discussion on the difference between using a DDPM-only model and VAE with DDPM prior. Basically, what would be the benefits of one over the other.

---

### Official Review · Reviewer_WPw4 · 2021-06-12

**Rating:** Borderline Accept
**Confidence:** 4

**Summary:**

This paper proposes using a diffusion process as the prior for a variational autoencoder (VAE).  Training is done via a lower bound on the ELBO (Equation 17).  Experimental results are reported for CelebA and CIFAR10, showing no conclusive improvements over a normalizing flow prior.

**Justification For Rating:**

I rated this paper as "borderline accept" because its proposed methodology doesn't involve a model with an explicit likelihood or an invertible NN.  Both the VAE and diffusion prior of interest require approximations to evaluate their likelihood.  Yet, the paper does use a normalizing flow as a baseline.

Furthermore, I found the paper to be conceptually weak since no strong motivation for the diffusion-based prior is given, other than it being a flexible model class of recent interest.  The paper briefly mentions the problem of prior vs agg. posterior mismatch (which motivates an expressive prior), but this is never examined in the experiments.

"We improve VAEs..." (abstract):  I found this misleading.  I was waiting for the paper to show some clear improvement but one never came, as the diffusion-based model never clearly improved upon the flow-based model.

---

### Decision · Program_Chairs · 2021-06-15

Accept (poster)